# Performance of two plasma separation devices for HIV-1 viral load measurement in primary healthcare settings

Adolfo Vubil,[1] Ana Flora Zicai,[1] Nádia Sitoe,[1] Carina Nhachigule,[1] Paulino da Costa,[1] Cacildo Magul,[1] Bindiya Meggi,[1] Sofia Viegas,[1] Nédio Mabunda,[1] Ilesh Jani[1]

**ABSTRACT** Dried blood spot (DBS) may overestimate the viral RNA, mainly in patients with low viral load (VL), due to proviral DNA and intracellular RNA. The Burnett and HemaSpot provide integrated solutions for the collection, separation, and drying of plasma from whole blood. This study aims to evaluate the performance of both devices compared to plasma to identify antiretroviral therapy (ART) failure. The devices were separately evaluated in a cross-sectional design. Patients on ART were included for the studies (Burnett: 611, October 2019 to January 2020) and (HemaSpot: 620, November 2020 to April 2021). VL was tested using CAP/CTM96. The sensitivity and specificity of DBS, Burnett, and HemaSpot were determined, and plasma results were considered as a reference at a threshold of 1,000 copies/ml. For the Burnett study, 2,444 specimens, including plasma, DBS, venous Burnett (vBurnett), and capillary Burnett (cBurnett), were collected. Sensitivity of DBS, vBurnett, and cBurnett was 97.4%, 98.3%, and 97.5%, respectively, whereas specificity was 86.8% for DBS, 96.9% for vBurnett, and 93.9% for cBurnett. For the HemaSpot study, 1,860 specimens were collected, including plasma, DBS, and vHemaSpot. Sensitivity of DBS and vHemaSpot was 95.0% and 91.3%, respectively, whereas specificity was 86.9% for DBS and 94.5% for vHemaSpot. The misclassification rate was more prominent in DBS (4.8%) and HemaSpot (8.4%) but lower in vBurnett (2.0%) and cBurnett (3.2%). The Burnett showed better performance than DBS, whereas HemaSpot showed poorer performance than DBS. Nevertheless, both Burnett and HemaSpot have high rate of non-reportable results. In the current format, neither of the two devices is feasible for VL scale-up in resource-limited settings.

**IMPORTANCE** Burnett and HemaSpot are two novel technologies that allow whole blood collection and plasma separation and stabilization at room temperature without the need of additional equipment. Hence, these devices are potential alternatives to fresh plasma as a suitable specimen for viral load scale-up to monitor antiretroviral therapy in resource-limited settings

**KEYWORDS** HIV-1 viral load, antiretroviral Therapy, primary healthcare

The monitoring of antiretroviral therapy (ART) is best achieved through the periodic measurement of HIV-1 viral load (VL) in plasma (1, 2). The collection, storage, and transport of fresh plasma to reference laboratories continue to pose logistical challenges in sub-Saharan Africa in general and particularly in Mozambique where 81.0% of patients was in treatment and 71.0% achieved viral suppression as of December 2021 (3). Dried blood spots (DBSs), which consist of whole blood dispensed onto filter paper cards, have been widely utilized as an alternative to fresh plasma (1, 2, 4–6). Filter card specimens can be obtained through finger or heel-pricks, do not require the use of ancillary equipment such as centrifuges, and importantly, can be stored and transported at room temperature.

Address correspondence to Adolfo Vubil, adolfo.vubil@ins.gov.mz.

The authors declare no conflict of interest.

The use of DBS has driven rapid scale-up of VL testing in high-burden resource-limited settings (1, 2, 4–8). Nevertheless, the measurement of VL in DBS can be inaccurate given that besides RNA present in plasma specimen, it includes proviral DNA and intracellular RNA which are present in whole blood. This may lead to an overestimation of VL, especially around the clinically relevant threshold of 1,000 copies/mL employed to identify patients with virological failure. Conversely, in most VL wide scale-up programs, no precision pipette is used to prepare DBS. So, the volume of whole blood in each of the five delineated areas of the DBS is not certain and is lower. This may lead to an underestimation of VL.

Incorrect clinical judgment based on erroneous VL values may lead to an unnecessary switch of ART regimens. Therefore, despite the acceptable analytical performance (8) and the fact that the use of DBS specimens increased access to VL testing in hard-to-reach settings, the limitations described above may have caused some analytical inaccuracy with substantial programmatic implications. Furthermore, additional specimen types and methodologies would be very useful to increasingly reduce the mean difference with plasma and have more choices and options for countries as they aim to scale up VL testing.

Recently, devices that collect whole blood onto membranes that concurrently separate plasma for analytical purposes have become available. These technologies have the logistical advantages of DBS, with the promise of better analytical performance for VL determination using plasma instead of whole blood. One such device is the cobas Plasma Separation Card that has shown to yield accurate VL results under field conditions in sub-Saharan Africa (9–11).

The Burnett Plasma Separation Device (12) and HemaSpot Plasma Separation Device are two novel technologies that allow whole blood collection and plasma separation and stabilization at room temperature without the need of additional equipment. Our study aimed to evaluate the performance of the Burnett device and HemaSpot device for HIV-1 VL determination in patients attending primary healthcare facilities in Mozambique.

## MATERIALS AND METHODS

### Study sites

Both Burnett and HemaSpot devices evaluations were conducted at *Primeiro de Maio* and *Polana Caniço* Health Centers in Maputo city, Mozambique. Both health centers provide general primary healthcare services, including ART. Routine VL for ART monitoring is performed using DBS specimens. Testing was done at the Instituto Nacional de Saúde (INS) laboratory, located less than 2 hours driving time from both health facilities. The INS is the national reference laboratory in Mozambique and performs VL assays that are ISO 15189 accredited.

### Study design and participants

Evaluations were conducted in a cross-sectional design. Performance for VL determination using Burnett and HemaSpot devices was investigated in 611 and 620 HIV-1-infected adult patients on ART, respectively. Patients for the study were consecutively enrolled at both health centers from October 2019 to January 2020 for the Burnett device, and from November 2020 to April 2021 for the HemaSpot device. Demographic and clinical information for each patient was collected employing a standardized data collection form.

HIV-1 VL was determined in fresh plasma, DBS, and the devices under evaluation for each patient. Both venous and capillary specimens were used for Burnett device, while for HemaSpot device, only venous specimens were available. Results obtained from DBS were reported back to patients as per the standard routine practice, whereas VL values in fresh plasma were used as a reference to assess the performance of the new devices. Laboratory testing was performed by blinded technicians who were trained and certified by the manufacturer of the VL test.

## Preparation of Burnett and HemaSpot plasma separation devices

The Burnett/VLPlasma (Nanjing BioPoint Diagnostics, Nanjing, China) and the HemaSpot (Spot on Sciences, Inc., Austin, TX, USA) are instrument-free plasma separator devices. Both technologies provide integrated solutions for the collection, separation, and drying of cell-free plasma from whole blood. These devices can be used for molecular and serological testing for various biomarkers.

The Burnett device is based on lateral flow chromatography and has a spiral-shaped form where the blood cells will be concentrated while allowing the flow of plasma.

The HemaSpot device is based on spiral flow chromatography and is coated with a nitrocellulose membrane and glass fiber which retain blood cells while allowing the plasma flow.

Evaluation of the performance of capillary blood for VL determination was done only for the Burnett device. For this purpose, a single-use safety lancet blue blade with a penetration depth of 2.0 mm was used for finger puncture. After the finger prick, a capillary tube marked 140 µL was filled, and blood was transferred onto the device. A full drop of phosphate-buffered saline provided in the kit was added after 3 min. The device was then stored at room temperature during 24 hours for drying.

To prepare the Burnett and HemaSpot devices with venous blood in the health facility, a capillary tube marked 140 µL was filled from a previously collected BD Vacutainer K2EDTA tube (Becton, Dickinson and Company, 1 Becton Drive, Franklin Lakes, NJ, USA) and transferred onto the devices. Both Burnett and HemaSpot specimens were prepared by the intended end-users in the healthcare facility and shipped to the INS laboratory for VL testing.

## Preparation of fresh plasma and DBSs

A total of 6.0 mL venous blood was collected from each patient in a BD Vacutainer K2EDTA tube. To prepare the DBS specimen at the health facility, a plastic Pasteur pipette was used to transfer one to two full drops of whole blood from the K2EDTA tube onto each of the five delineated areas in the DBS card (Ahlstrom, Germany GmbH). Whole blood from the K2EDTA tube was also utilized to prepare Burnett and HemaSpot devices as described above. The remnant anticoagulated whole blood was transported to INS laboratories within 6 hours post-venipuncture for plasma separation and storage at −80°C until VL testing.

After drying, DBS, capillary Burnett, venous Burnett, and venous HemaSpot specimens were packaged in separate gas impermeable zip-lock bags with desiccant and subsequently shipped at room temperature to INS laboratories. All Burnett and HemaSpot devices were stored at room temperature and tested within 72 hours after collection. Fresh plasma specimens were tested in parallel with Burnett and HemaSpot, whereas DBS took priority on testing as part of the routine clinical management of patients.

## Pre-testing elution of specimens

One spot of DBS was eluted in 1,000 µL of phosphate-buffered saline, pH 7.4 (1×; Thermofisher Scientific, USA) and incubated for 30 min at room temperature (18°C–25°C). After the incubation, specimens were manually homogenized and immediately loaded into the CAP/CTM 96 v2 (Roche Molecular Diagnostics, Branchburg NJ, USA) for testing using the free virus elution Roche protocol, as per manufacturer's recommendations.

For Burnett and HemaSpot specimens, each plasma spot was eluted in 1,100 µL of Sample Pre-Extraction (Spex) solution (Roche Systems, Pleasanton, CA, USA) and incubated at 56°C and 1,000 rpm for 10 min in a thermomixer. The HemaSpot specimens were incubated in a secondary tube. Subsequently, all 1,100 µL of supernatant were transferred to the testing tube (S-tube) to avoid the disintegration of the card and consequent pipetting obstruction in the automated platform. The Burnett specimens were incubated in the testing tube (S-tube) as this card does not disintegrate easily.

After incubation, specimens were immediately loaded into CAP/CTM 96 v2 automated platform for testing.

Fresh plasma was separated from whole blood specimens through centrifugation at 800–1,600 g for 20 min at room temperature (18°C–25°C) and stored at −80°C up to testing day. A total of 1,100 µL of fresh plasma was used for VL testing in the CAP/CTM 96 v2 platform.

## HIV-1 VL testing

VL measurement for all specimen types was conducted using the CAP/CTM 96 HIV-1 Quantitative Test v2 (Roche Molecular Diagnostics, Branchburg NJ, USA), according to the manufacturer's instructions. The test definition files used were HI2PSC96 for Burnett and HemaSpot, IFS96CDC for DBS, and HI2CAP96 for fresh plasma specimens.

Fresh plasma and DBS specimens that failed to report VL results at the first testing were re-tested whenever specimens were available. The Burnett and HemaSpot specimens which failed to report results at the first testing were not re-tested, as these devices are for single testing only.

The interpretation of VL results followed the manufacturer's instructions, which establish 20 and 400 copies/mL of viral RNA as the Low Limit of Quantification (LLoQ) for plasma and DBS specimens, respectively. There is no specific LLoQ defined for the Burnett and HemaSpot specimens. For these specimens, an LLoQ of 738 copies/mL was considered similar to the cobas Plasma Separation Card which has similar characteristics as Burnett and HemaSpot specimens and was previously evaluated using the same testing platform (CAP/CTM 96 HIV-1 Quantitative Test v2).

## Statistical analysis

All VL results were log10 transformed. Specimens with non-detectable VL results were assigned a value of 1 copy/mL. Specimens with values below the LLoQ for fresh plasma, DBS, Burnett, and HemaSpot specimens, were assigned a fixed value of 19, 390, and 737 copies/mL, respectively, to enable log10 transformation. Analyses were performed using STATA 14.2 (StataCorp, Texas, USA) and MedCalc Statistical Software version 16.4.3 (MedCalc Software bvba, Ostend, Belgium). For categorical variables, descriptive analysis was performed, and the data were summarized in proportions and frequency table.

Sensitivity, specificity, positive and negative predictive values, and misclassification rates of DBS, Burnett, and HemaSpot were calculated against a threshold of 1,000 copies/mL (13, 14), taking VL results obtained in fresh plasma as reference (15). The misclassification rate corresponds to the proportion of VL values in Burnett or HemaSpot specimens with discordant classification to that of plasma at the threshold of 1,000 copies/mL. For the present study, the general rate was calculated and then disaggregated into false positive which correspond to VL values above 1,000 copies/mL in Burnett and HemaSpot specimens, whereas in plasma specimens, the VL values are below 1,000 copies/mL. The false negatives correspond to VL values below 1,000 copies/mL in Burnett and HemaSpot specimens, whereas in plasma specimens the values are above 1,000 copies/mL. This was calculated using the Rsdtudio statistic package. Concordance correlation (16) and Bland-Altman (17) analyses were performed to determine the agreement of VL values generated by the various methods. Only VL results ≥1,000 copies/mL were included in these latter analyses.

## RESULTS

### Clinical and demographic characteristics of patients

Patients under routine ART monitoring in Primeiro de Maio and Polana Caniço Health Centers were consecutively enrolled for evaluating the performance of Burnett and HemaSpot devices for HIV-1 VL determination. Study participants were mostly in the 25–54 years age groups, with the majority being female (Table 1). Age median (interquartile range[IQR]) of 41 (18–84) years and 40 (18–83) years were observed for Burnett

**TABLE 1** Demographic and clinical characteristics of study participants[a]

| | | Burnett device (N = 611) | | HemaSpot device (N = 620) | |
|---|---|---|---|---|---|
| | | n | % | n | % |
| Sex | Male | 222 | 36.3 | 212 | 34.2 |
| | Female | 372 | 60.9 | 386 | 62.3 |
| | Not available | 17 | 2.8 | 22 | 3.5 |
| Age (years) | 15–24 | 28 | 4.5 | 31 | 5.0 |
| | 25–34 | 129 | 21.1 | 133 | 21.4 |
| | 35–44 | 212 | 34.7 | 144 | 39.3 |
| | 45–54 | 120 | 19.6 | 122 | 19.6 |
| | ≥55 | 87 | 14.2 | 79 | 12.7 |
| Plasma VL (copies/mL) | Not detected | 279 | 45.7 | 336 | 54.2 |
| | <20 | 76 | 12.4 | 88 | 14.2 |
| | 21–1,000 | 131 | 21.4 | 86 | 13.9 |
| | 1,001–10,000 | 20 | 3.2 | 22 | 3.5 |
| | 10,001–100,000 | 56 | 9.2 | 34 | 5.5 |
| | >100,001 | 45 | 7.3 | 42 | 6.8 |
| | Missing | 4 | 0.7 | 12 | 1.9 |
| Reason for VL testing | ART routine monitoring | 532 | 87.1 | 442 | 71.3 |
| | Treatment failure suspicion | 64 | 10.5 | 22 | 3.5 |
| | Not available | 15 | 2.5 | 156 | 25.2 |

[a]ART - antiretroviral therapy; VL - viral load; TDF - Tenofovir; 3TC - Lamivudine; EFV - Efavirenz, LPV/r - Lopinavir/ritonavir.

and HemaSpot studies, respectively. Over three-quarters of patients were receiving first line scheme of ART based on tenofovir combined with lamivudine and efavirenz. About half of the study participants had undetectable plasma VL, with only a small proportion having VL higher than 10,000 copies/mL. VL median (IQR) of 100 (20–33,085) copies/mL and 120 (20–17,341) copies/mL were observed for Burnett and HemaSpot studies, respectively.

## Performance of the Burnett plasma separation device

From the 611 patients enrolled in the Burnett study, a total of 2,444 specimens were collected, including fresh plasma, DBS, venous, and capillary Burnett devices. Reportable results were generated in 85.2% (2,082/2,444) specimens, including 77.1% (471/611) fresh plasma, 92.5% (565/611) capillary Burnett, 89.0% (544/611) venous Burnett, and 82.2% (502/611) DBS specimens after first testing. After re-testing, the general reportable results rate increased to 94.7% (2,315/2,444), 99.5% (608/611) for fresh plasma, and 99.3% (607/611) for DBS specimens. The Burnett and HemaSpot specimens were not re-tested due to a lack of specimen. In this study, 4.9% (120/2,444) specimens, which included 0.49% (3/611) fresh plasma, 0.65% (4/611) DBS, 10.1% (67/611) venous Burnett, and 7.5% (46/611) capillary Burnett specimens definitely failed to report results (Fig. 1).

From the 2,315 specimens with reported results, 498 patients had results for all four sample types: fresh plasma, DBS, venous Burnett, and capillary Burnett specimens.

The performance of the Burnett device to identify viral suppression at the clinically relevant threshold of 1,000 copies/mL was investigated by measuring the sensitivity, specificity, and misclassification rate against fresh plasma as outlined in Table 2. The overall sensitivity of the Burnett prepared using venous blood or capillary blood was 98.3% (97.0–99.5) and 97.5% (96.0–99.0), respectively, which were not significantly better than the 97.4% (95.9–99.0) of the DBS, as shown by overlapping confidence intervals. The specificity of the Burnett device prepared using venous blood or capillary blood was 96.9% (93.5–100) and 93.9% (89.1–98.6), also not significantly different from the 86.8%

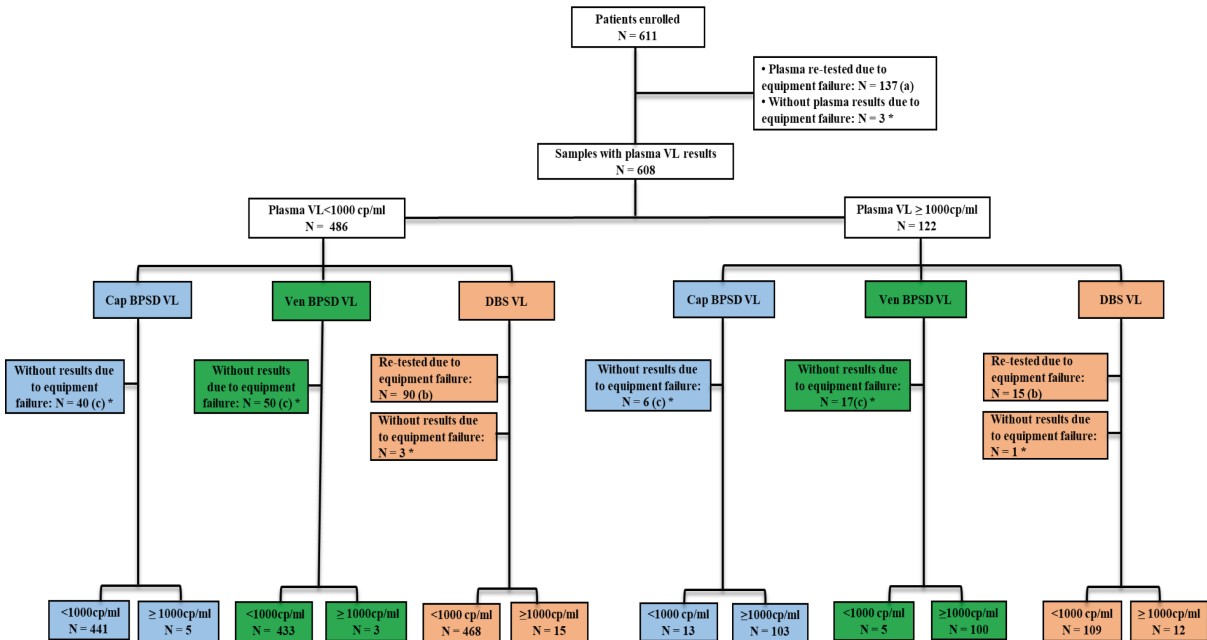

**FIG 1** Flow diagram illustrates the total number of patients recruited for evaluation of the Burnett device, number of samples collected for each specimen type, results failed and reported for each specimen type, and the total number of patients with result for the four specimen types. (A) Eleven samples failed due to Quantitation Standard (QS) invalid flag remark and 126 samples failed due to controls failure; (B) 99 samples failed due to controls failure, 2 samples failed due to a small volume of specimen detected by the equipment during testing, 1 due to sample clot, and 3 due to QS_invalid flag remark; (C) 39 samples failed due to controls failure and 7 samples failed due to QS_invalid flag remark; (D) 61 samples failed due to controls failure, 1 due to sample clot, 4 due to QS_invalid, and 1 due to a small volume of specimen detected; *not re-tested due to a lack of specimen. VL – viral load; Cap BPSD – Capillary Burnett Plasma Separation Device; Ven BPSD – Venous Burnett Plasma Separation Device; DBS – Dried Blood Spot.

(80.3–93.2) for DBS. However, the misclassification rate for DBS specimens was 4.8%, higher than the 2.0% and 3.2% for venous and capillary Burnett specimens, respectively.

The concordance correlation coefficients between VL values obtained in fresh plasma and in Burnett device venous and capillary specimens were 0.92 and 0.91, respectively, which were higher when compared to 0.86 for DBS specimens. Moreover, limits of agreement for both venous (−0.952 to +1.137) and capillary (−1.008 to +1.361) Burnett specimens were narrower than those observed for DBS (−1.341 to +1.532; Fig. 2).

## Performance of the HemaSpot plasma separation device

A total of 1,860 specimens were collected from 620 patients, including fresh plasma, DBS, and venous HemaSpot. Reportable results were generated in 95.3% (1,772/1,860) specimens, which included 96.6% (599/620) fresh plasma, 91.8% (569/620) venous

**TABLE 2** Analytical performance of DBS, venous Burnett, capillary Burnett, and venous HemaSpot specimens using plasma as the gold standard to determine HIV-1 viral load[a]

| Variables | Burnett (CI 95%) | | | HemaSpot (CI 95%) | |
| --- | --- | --- | --- | --- | --- |
| | DBS | Venous | Capillary | DBS | Venous |
| Sensitivity[b] | 97.4 (95.9–99.0) | 98.3 (97.0–99.5) | 97.5 (96.0–99.0) | 95.0 (93.1–96.9) | 91.3 (88.9–93.7) |
| Specificity[b] | 86.8 (80.3–93.2) | 96.9 (93.5–100.3) | 93.9 (89.1–98.6) | 86.9 (79.7–94.1) | 94.5 (88.5–100.5) |
| PPV[b] | 96.5 (94.6–98.3) | 99.2 (98.4–100.1) | 98.5 (97.3–99.7) | 97.7 (96.4–99.1) | 99.4 (98.7–100.1) |
| NPV[b] | 90.2 (84.4–96.0) | 93.1 (88.2–98.0) | 90.2 (84.4–96.0) | 74.5 (65.9–83.1) | 53.1 (43.2–62.9) |
| Misclassification rate[b] | 4.8 (3.1–7.1) | 2.0 (1.0–3.7) | 3.2 (1.8–5.2) | 6.2 (4.4–8.4) | 8.4 (6.3–10.9) |
| False positive rate[b] | 9.8 (5.1–17.7) | 6.9 (3.0–14.1) | 9.8 (5.1–17.7) | 25.5 (17.5–35.5) | 46.9 (36.9–57.2) |
| False negative rate[b] | 3.5 (2.0–6.0) | 0.8 (0.2–2.4) | 1.5 (0.6–3.4) | 2.3 (1.2–4.1) | 0.6 (0.16–1.9) |

[a]DBS – dried blood spot; PPV – positive predictive value; NPV– negative predictive value; CI – confidence interval.
[b]To detect plasma value of 1,000 copies/mL.

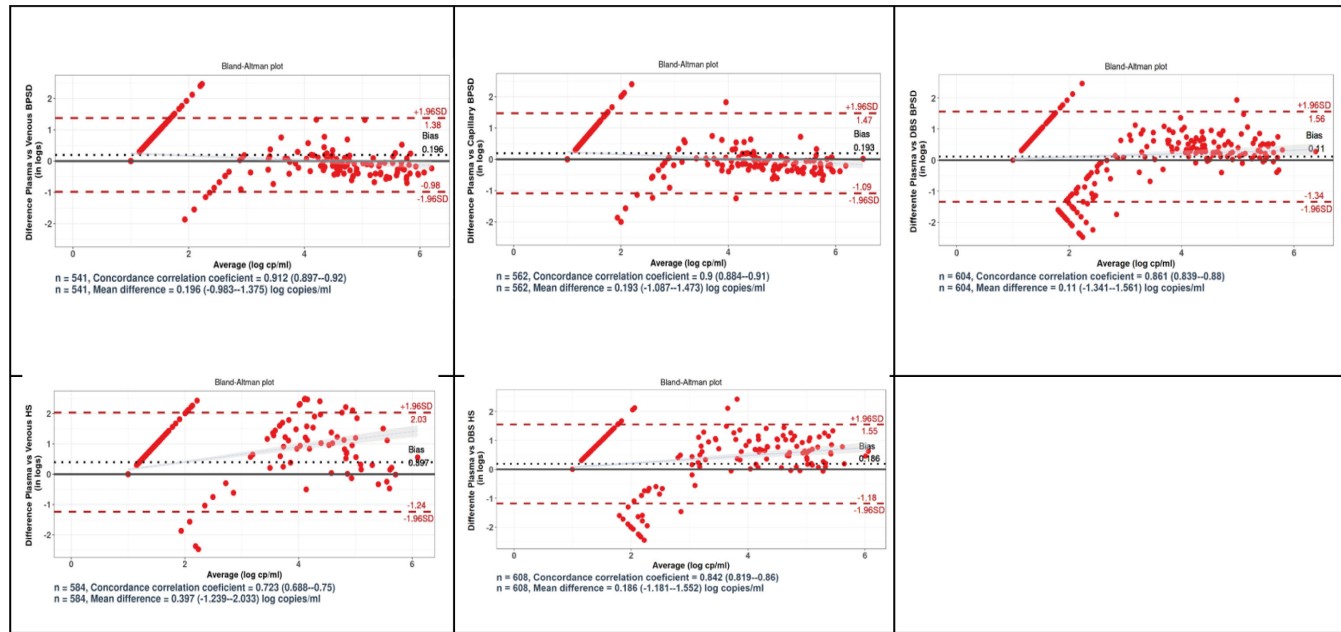

**FIG 2** Scatter plots of absolute difference in log copies/mL VL vs mean VL (Bland-Altman analysis). The vertical axis is the log difference between test and reference. The bias (in dotted black lines) and LOA (Limits of Agreement in dotted red lines) are indicated on the plots. The legend also includes the number of specimens used for Bland Altman analysis and calculation of concordance correlation coefficients. DBS HS (DBS specimens used for the HemaSpot device evaluation). DBS BPSD (DBS specimens used for the Burnett device evaluation). The straight diagonal lines in the graphic correspond to specimens with non-detectable VL results which were assigned a value of 1 copy/mL.

HemaSpot, and 97.4% (604/620) DBS specimen after first testing. After re-testing, the reportable results rate increased to 95.9% (1,783/1,860) specimens. For this study, 4.1% (77/1,860), including 1.94% (12/620) fresh plasma, 2.26% (14/620) DBS, and 8.2% (51/620) venous HemaSpot specimens failed to report results after re-testing (Fig. 3). From the 1,759 specimens with reported results, 550 patients had paired results for all specimens: fresh plasma, DBS, and venous HemaSpot.

The performance of HemaSpot to identify viral suppression at the clinically relevant threshold of 1,000 copies/mL as determined by fresh plasma was investigated by measuring the sensitivity, specificity, and misclassification rate as outlined in Table 2. The overall sensitivity of venous HemaSpot was 91.3% (88.9–93.7) to identify patients with VL below 1,000 copies/mL, not significantly better than the sensitivity for DBS, 95.0% (93.1–96.9), as shown by overlapping confidence intervals. Although the specificity for venous HemaSpot was higher at 94.5% (88.5–100.5), it was not significantly different from the sensitivity for DBS at 86.9% (79.7–94.1). The HemaSpot device showed a higher misclassification rate at 8.4% compared to 6.2% for DBS.

The concordance correlation coefficient between VL values obtained from paired fresh plasma and venous HemaSpot (0.72) was lower compared to 0.84 for DBS. Moreover, limits of agreement for venous HemaSpot (−1.24 to +2.03) were wider than those observed for DBS (−1.19 to +1.58; Fig. 2).

## DISCUSSION

Timely access to accurate HIV-1 VL testing remains problematic for millions of patients undergoing ART in sub-Saharan Africa. Programmatic challenges with the use of fresh plasma led many countries to adopt DBS specimens for HIV-1 VL testing (1, 5, 7, 8), even though these have been shown to yield inaccurate results (1, 9, 11, 18). We evaluated two novel specimen collection devices that have the potential to generate accurate VL measurements while keeping the programmatic advantages of DBS.

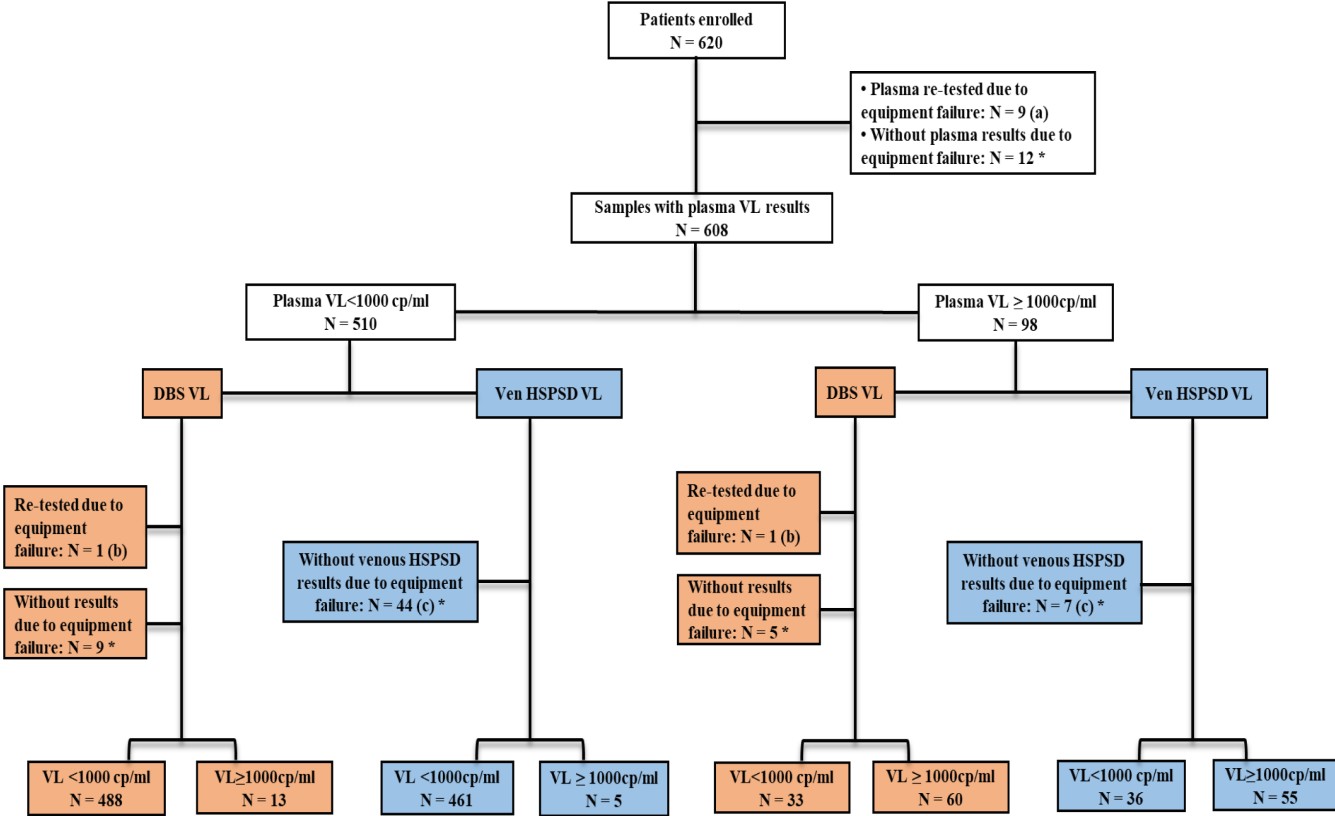

**FIG 3** Flow diagram of patients recruited and reported viral load results using fresh plasma, DBS, and venous HemaSpot samples. The diagram shows the total number of results failed and the total number of patients with reported results for the three specimen types. (A) Four samples failed due to sample clot flag remark, five due to instrument troubleshoots, and three due to Quantitation Standard (QS)_invalid flag remark; (B) two samples failed due to sample clot flag remark; (C) 40 samples failed due to sample clot, 8 due to QS_invalid, and 3 due to a small volume of specimen detected by the equipment during testing; *not re-tested due to lack of specimen. VL – viral load; Ven HSPSD – Venous HemaSpot Plasma Separation Device; DBS – Dried Blood Spot.

In our study, 10.1% of venous Burnett, 7.5% of capillary Burnett, and 8.7% of venous HemaSpot specimens definitively failed to report results. The high rates of non-reportable results were a consequence of the inability to re-test in case of invalid results as both devices collect a sole blood spot. This constitutes a major limitation for their routine use in resource-limited health systems, which are frequently affected by issues such as failures of electricity supply, breakdown of testing equipment, and specimens with inadequate quality (7, 8, 19–21). The low proportion of reportable results for the plasma specimens observed after first testing in the Burnett study is related to some electricity issues that occurred in the laboratory during the period of plasma specimen testing.

When valid results only were taken into consideration, the Burnett specimen showed an overall better performance than DBS, with equivalent sensitivity and specificity but with lower misclassification rates and narrower limits of agreement. This result is similar to previous data from a study conducted in Malaysia (12). Our data show that Burnett prepared either from capillary or venous blood specimens generate the same trend of VL results as plasma specimens. This is because the Burnett specimen eliminates the over-quantification of HIV-1 nucleic acid, through the elimination of cell-associated RNA molecules and proviral DNA interference. Conversely, the HemaSpot specimen showed an overall poorer performance than DBS, with equivalent sensitivity and specificity but with higher misclassification rates and wider limits of agreement. This shows that HemaSpot specimens over-quantify the true HIV VL as measured by the reference plasma specimen and cannot constitute an alternative to fresh plasma for VL testing in the current format. Additionally, it was expected that HemaSpot specimen could eliminate

the interference of proviral DNA on the VL results. However, the high rate of false positive results observed in our study suggests a low efficiency of this device to separate plasma. This factor may impact the measurement of VL mainly around the clinically relevant threshold of 1,000 copies/mL.

Our findings contrast with those yielded for the cobas Plasma Separation Card, which showed an analytical performance superior to DBS and equivalent to liquid plasma in South Africa and Mozambique (9, 11, 13, 21). Technical characteristics that contribute to a better performance of the cobas Plasma Separation Card include the porous membrane which allows only plasma to filter through while retaining all other blood components.

As the number of people undergoing ART in resource-poor settings continues to grow, it becomes increasingly urgent to identify technologies that allow accurate and timely measurement of HIV-1 VL at the primary healthcare level. Innovative collection and transport methods, such as the Burnett and HemaSpot devices evaluated here, could play an important role in increasing access to quality VL testing and could represent a choice and option for VL scale-up in resource-limited settings. Of the two specimen types evaluated, the Burnett offers the most promise if the shortcoming posed by the collection of a single blood spot could be overcome. The current format of both Burnett and HemaSpot devices could be adjusted for at least three spots or by collecting two or three devices per patient to deal with the frequent test failures caused mainly by equipment breakdown and electricity issues. This approach is used in most of VL programs in sub-Saharan Africa. In the current format, however, neither of the two devices is suitable for widescale implementation in sub-Saharan Africa.

Some of the limitation of this study includes the lack of capillary HemaSpot specimen for evaluation and the lack of backup specimen for re-testing in case of failure for both Burnett and HemaSpot devices.

## ACKNOWLEDGMENTS

We would like to gratefully acknowledge the laboratory staff and nurses at the Primeiro de Maio and Polana Caniço Health facilities, Maputo, Mozambique, for specimen collection. We acknowledge the Bill and Melinda Gates Foundation for funding opportunity #OPP1171455.

A.V.: Designed study, oversaw data and specimen collection, oversaw specimen testing, interpreted results, wrote the first draft, and approved the finalized version of the manuscript. A.Z. and C.N.: Designed study, collected specimens and data, conducted specimen testing, interpreted results, and commented on the manuscript drafts. N.S., B.M., N.M., and S.V.: Designed study, interpreted results, commented on the manuscript drafts, and approved the finalized version of the manuscript. O.L.: Conducted data analysis, interpreted results, commented on the manuscript drafts, and approved the finalized version of the manuscript. P.C.: Conducted data analysis, interpreted results, and produced the finalized version of the manuscript. I.J.: Acquisition of the financial support for the project leading to this publication, designed study, interpreted results, wrote the first draft, and produced the finalized version of the manuscript.

## AUTHOR AFFILIATION

[1]Laboratory Reference Service, Instituto Nacional de Saúde, Marracuene, Mozambique

## AUTHOR ORCIDs

Adolfo Vubil  http://orcid.org/0000-0002-1948-037X

## ETHICS APPROVAL

Ethical approval for the study was attained from Mozambique's National Health Bioethics Committee in March 2019, with the reference number 136/CNBS. Written informed

consent was obtained from each participant prior to conducting any study procedure. Patients were given a copy of the signed consent form, which contained information about the study as well as contacts details of the principal investigator and the ethics committee.

## ADDITIONAL FILES

The following material is available online.

### Open Peer Review

**PEER REVIEW HISTORY (review-history.pdf).** An accounting of the reviewer comments and feedback.

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
