## [Reviewer comments · Microbiology Spectrum]

Microbiology Spectrum

Performance of Two Plasma Separation Devices for HIV Viral Load Measurement in Primary Healthcare Settings

Adolfo Vubil, Ana Flora Zicai, Nádia Siteo, Carina Nhachigule, Paulino Da Costa, Cacildo Magul, Bindiya Meggi, Sofia Viegas, Nédio Mabunda, and Ilesh Jani

Corresponding Author(s): Adolfo Vubil, Instituto Nacional de Saúde-Mozambique

Review Timeline:

Submission Date:	February 5, 2023
Editorial Decision:	March 23, 2023
Revision Received:	July 28, 2023
Accepted:	August 25, 2023

Editor: Yongjun Sui

Reviewer(s): The reviewers have opted to remain anonymous.

Transaction Report:

DOI: <https://doi.org/10.1128/spectrum.00546-23>

March 23, 2023

Dr. Adolfo Vubil
Instituto Nacional de Saúde-Mozambique
Estrada Nacional número 1
Marracuene-Maputo
Mozambique

Re: Spectrum00546-23 (Performance of Two Plasma Separation Devices for HIV Viral Load Measurement in Primary Healthcare Settings)

Dear Dr. Adolfo Vubil:

Link Not Available

Sincerely,

Yongjun Sui

Journals Department
Reviewer comments:

Reviewer #1 (Comments for the Author):

This study looked at the diagnostics accuracy of two new plasma separation devices for HIV viral load testing. These sample collection technologies add to those currently available that may allow for increased access to viral load in resource-limited settings. The manuscript is solid, but could be improved with some consideration of the below comments.

Major comments

1. The conclusion of this study - that neither device is feasible - was unclear to follow. The results, for the most part, seem to be similar or better than those provided by DBS, either in this study or the international data; however, the authors suggest this isn't

acceptable. DBS is used widely and has been determined by the global community to perform at a level acceptable enough given the improved access it provides. Why these technologies wouldn't be useful given the performance results provided is unclear to the reader (besides perhaps the challenge of one specimen only).

2. It would be very beneficial to explain these technologies a bit more. It was hard to envision whether they are like the Roche plasma separation card or something else, especially given the authors suggest that one of them measures intracellular NATs (line 370 and 376-379; then they aren't a 'plasma separation' device). For example, what are the steps needed by the health care worker, what is the output post-blood application, etc. Further, on line 184, it would be helpful to know whether plasma or whole blood is eluted from the spots, particularly given that SPEX is used (which we know is too rough for whole blood and leads to considerable overquantification).

3. To the point above, line 94 is incorrect. DBS do not measure plasma VL. DBS measure plasma as well as intracellular viral loads. The latter portion includes proviral DNA and intracellular RNA. Further, the authors suggest that DBS are inaccurate (introduction and discussion). It might be worth reviewing some of the more recent literature and guidance on this, as the field now generally accepts the performance of DBS given the increased access to testing that it provides. Finally, in the following sentences the authors suggest only an overestimation. It might be worth also noting that DBS could suffer from underestimation given lower volumes of sample used.

4. Following the above issues in the abstract, the authors would improve their introduction with some discussion on viral load in the context of TLD (lines 95-101) and acknowledging the current misclassification rates seen with DBS (review recent PLoS Medicine meta-analysis) showing limited misclassification that would result in incorrect clinical decisions. All of this together to say, I don't think this manuscript needs to be so negative on DBS to justify the consideration of additional sample types. It would be very useful to have more choice and options for countries as they aim to scale up viral load.

5. It would be helpful to include in the methods (lines ~150) whether the plasma separation devices were all prepared in the lab or in the health care facility by the intended end-users before shipping. I.e. was this a laboratory evaluation or field evaluation?

6. It should be clarified in the results of the Abstract that the performance stated is in comparison to plasma.

7. It would be helpful to define in the methods how misclassification is calculated. Is it overall or upward or downward? And what was the formula used. Further, the false positive and false negative rates are close but not quite 1-specificity and 1-sensitivity, respectively. Please either include how these were calculated and/or confirm the values.

8. The conclusions touch on a high rate of non-reportable results, but this seems more an implementation challenge. Is there a suggested rate that all technologies should meet? What would be the consequence of collecting more than one sample from each participant to help resolve this issue (costs, feasibility)? Can the suppliers adjust the technology to include more spots (this is unclear for the reader because little is known about these devices)?

9. In the statistical analysis of the methods, the authors should seriously reconsider how they handle certain results. In particular, undetectable results are absolutely not the same as results that are detectable but below the LoQ. The latter should not be considered undetectable as they represent samples with low levels of viremia. Instead, they could be given the value of one viral load less than the LoQ. This seems to be different in Table 1 that shows a differentiation between those not detected and those <20 copies/ml. Some clarity here would be good.

10. In Table 1, it isn't clear what the median represents, especially for the HemaSpot that has an 'X'.

11. The proportion of reportable results for fresh plasma within the Burnett study is very low (77.1%). It would be useful to describe why this might have occurred. Further when indicating 'after re-testing', it would be useful to describe with which technologies (all or only plasma and DBS were re-tested).

12. The figures are currently presented as Figure 1, 3, and 2. This should be re-ordered in sequence of presentation in the results.

13. In the paragraph starting line 307, the proportion of reportable results started as 95.3% and then decreased to 94.6%; however, this was presented as an increase. Please confirm the values.

14. In Figure 3, it would be helpful to explain the straight diagonal line of results in each graph (some graphs have multiple). Further the y-axis could be improved (ie. what is 'DBS BPSD', I thought 'normal' DBS cards were used).

Minor comments

1. Line 19: 'provide' should be singular.

2. Line 29: add 'the' before Burnett.

3. Line 95 seems to have a footnote indicated, but no footnote.

4. Replace 'ART schemes' with 'ART regimens'.

5. Line 106, replace 'measurement' with 'results'. Also here it would be useful to provide the high level results from those studies of the PSC.

6. Line 111, remove 'device' after Burnett and make it plural on line 112. Similar on line 118, remove 'device' after Burnett.

7. Line 124, is the assay or lab ISO accredited?

8. Line 135, 'the' Burnett device.

9. Line 142, unclear what the 'SETM/HS' means - it doesn't seem to be included elsewhere.

10. Line 280, I would suggest '498 patients had results for all four sample types: plasma...'

11. Similar to above, line 315 could say '550 patients had paired results for all specimens: plasma...'

12. Line 322, 'below' is spelled incorrectly.

13. Line 322, instead of 'one' I would suggest writing out 'sensitivity' instead.

14. In the figures, 'clot' is misspelled.

Staff Comments:

Preparing Revision Guidelines

Please return the manuscript within 60 days; if you cannot complete the modification within this time period, please contact me. If you do not wish to modify the manuscript and prefer to submit it to another journal, please notify me of your decision immediately so that the manuscript may be formally withdrawn from consideration by Microbiology Spectrum.

Reviewer's comments

Performance of Two Plasma Separation Devices for HIV Viral Load Measurement in Primary Healthcare Settings

General comment

Specify the HIV type of focus. HIV-1 or HIV-2

Abstract

Line 20: Replace et with at

Background

General comment

Consider highlighting the disadvantages of using DBS in low VL

Provide more context about the current state of ART and VL monitoring in Mozambique specifically, as this study is being conducted in that setting

Line 95: Check the reference style and consider revising. One appears as a superscript

Study sites

Line 123-124 indicates that the VL assays are ISO accredited instead of the reference lab. Kindly rephrase this statement as appropriate.

HIV viral load testing

Line 211 mentions the cobas® Plasma Separation Card yet upto that point it does not appear in the methodology. Kindly clarify the relevance of its mention or inclusion.

Data analysis

The section does not provide information on how the socio demographic and clinical information of the study population was analyzed. Kindly include

Results

Line 237-242 reports on analyzed socio demographic and clinical information yet the methodology section does not provide information on how these data were analyzed. Kindly revise.

Table 1

The cumulative percentage for the Sex variable exceeds 100%.

The Median (IQR) results when presented on the table are confusing, consider presenting them on a paragraph

Figure 1

The legend is unclear. For example "2 samples failed due to no sample". Kindly rephrase

Consider renumbering your figures on the text because figure 3 comes before figure 2.

Line 286-290 represent significant findings. Kindly present the level of significance between the platforms using p values.

Table 2

The misclassification rate and false positive rate results of the HemaSpot are misrepresented. Kindly revise

Discussion

Line 364-365,367-368 indicate that a test platform was compared to a sample type. Kindly revise this section.

Line 370-372 mentions the advantages of the Burnett device while those of the HemaSpot device are not mentioned. Kindly revise.

The limitations of the study are missing. Kindly revise.

A lot of analytics have been done and presented. However, the meaning and interpretation of these finding has not been explained. Kindly consider revising

Reviewer comments:

Reviewer #1 (Comments for the Author):

This study looked at the diagnostics accuracy of two new plasma separation devices for HIV viral load testing. These sample collection technologies add to those currently available that may allow for increased access to viral load in resource-limited settings. The manuscript is solid, but could be improved with some consideration of the below comments.

Major comments

1. The conclusion of this study - that neither device is feasible - was unclear to follow. The results, for the most part, seem to be similar or better than those provided by DBS, either in this study or the international data; however, the authors suggest this isn't acceptable. DBS is used widely and has been determined by the global community to perform at a level acceptable enough given the improved access it provides. Why these technologies wouldn't be useful given the performance results provided is unclear to the reader (besides perhaps the challenge of one specimen only).

R: We agree with the good performance of the devices evaluated in this study (lines 391-394) and we mentioned the high rate of non-reportable results as a major limitation (lines 385-388). So, our conclusion was based not only on the analytical performance, but also on the high rate of samples without conclusive results (>7.0%) for both Burnett and HemaSpot specimens. We understand that this aspect will impact the proportion of patients without VL results for ART monitoring and this is crucial for decision on the adoption of a new device for widescale implementation.

2. It would be very beneficial to explain these technologies a bit more. It was hard to envision whether they are like the Roche plasma separation card or something else, especially given the authors suggest that one of them measures intracellular NATs (line 370 and 376-379; then they aren't a 'plasma separation' device). For example, what are the steps needed by the health care worker, what is the output post-blood application, etc. Further, on line 184, it would be helpful to know whether plasma or whole blood is eluted from the spots, particularly given that SPEX is used (which we know is too rough for whole blood and leads to considerable overquantification).

R: We agree with your comment. We provided more details about the technologies (lines 141 - 145). Furthermore, details about the steps needed by the health care workers for specimen preparation can be found in lines 147-151. Additionally, was clarified that plasma spot was eluted for both Burnett and HemaSpot specimens (line 182).

3. To the point above, line 94 is incorrect. DBS do not measure plasma VL. DBS measure plasma as well as intracellular viral loads. The latter portion includes proviral DNA and intracellular

RNA. Further, the authors suggest that DBS are inaccurate (introduction and discussion). It might be worth reviewing some of the more recent literature and guidance on this, as the field now generally accepts the performance of DBS given the increased access to testing that it provides. Finally, in the following sentences the authors suggest only an overestimation. It might be worth also noting that DBS could suffer from underestimation given lower volumes of sample used.

R: We agree with your comment. We reformulated the sentence and we think that now it's more clear that DBS measure plasma as well as intracellular viral load (lines 79- 81). Furthermore, we included a paragraph to clarify that viral load in DBS specimen can be affected as well by the lower volume of whole blood used (lines 83-86). Regarding the inaccuracy of the VL in DBS specimens, we agree that DBS leads to increased access to VL; however, we think that it is important to consider both access and analytical accuracy (lines 87-94).

4. Following the above issues in the abstract, the authors would improve their introduction with some discussion on viral load in the context of TLD (lines 95-101) and acknowledging the current misclassification rates seen with DBS (review recent PLoS Medicine meta-analysis) showing limited misclassification that would result in incorrect clinical decisions. All of this together to say, I don't think this manuscript needs to be so negative on DBS to justify the consideration of additional sample types. It would be very useful to have more choice and options for countries as they aim to scale up viral load.

R: We agree with your comment. We reformulated the introduction (lines 88-94).

5. It would be helpful to include in the methods (lines ~150) whether the plasma separation devices were all prepared in the lab or in the health care facility by the intended end-users before shipping. I.e. was this a laboratory evaluation or field evaluation?

R: Both Burnett and HemaSpot specimens were prepared at the health facilities by the intended end-user. This was made clearer in the methodology (lines 155-157).

6. It should be clarified in the results of the Abstract that the performance stated is in comparison to plasma.

R: This was clarified as suggested (line 5).

7. It would be helpful to define in the methods how misclassification is calculated. Is it overall or upward or downward? And what was the formula used. Further, the false positive and false negative rates are close but not quite 1-specificity and 1-sensitivity, respectively. Please either include how these were calculated and/or confirm the values.

R: The false positive and negative values were confirmed. Detailed explanation about how the misclassification rate was calculated was incorporated (lines 224-232).

8. The conclusions touch on a high rate of non-reportable results, but this seems more an

implementation challenge. Is there a suggested rate that all technologies should meet? What would be the consequence of collecting more than one sample from each participant to help resolve this issue (costs, feasibility)? Can the suppliers adjust the technology to include more spots (this is unclear for the reader because little is known about these devices)?

R: There are no suggested rates of non-reportable results for VL technologies. However, most of VL programs in sub-Saharan Africa recommends enough specimen for at least three tests given the frequent test failures caused mainly by equipment breakdown and electricity issues. This was incorporated in the discussion section (lines 421-426).

9. In the statistical analysis of the methods, the authors should seriously reconsider how they handle certain results. In particular, undetectable results are absolutely not the same as results that are detectable but below the LoQ. The latter should not be considered undetectable as they represent samples with low levels of viremia. Instead, they could be given the value of one viral load less than the LoQ. This seems to be different in Table 1 that shows a differentiation between those not detected and those <20 copies/ml. Some clarity here would be good.

R: We agree with your comment. We have revised the statistical analysis. Specimens with non-detectable VL results were assigned a value of 1 copy/ml whereas specimens with values below the LoQ were assigned a value of 19, 390 and 737 copies/ml for fresh plasma, DBS, Burnett and HemaSpot specimens respectively (lines 216-218).

10. In Table 1, it isn't clear what the median represents, especially for the HemaSpot that has an 'X'.

R: This was corrected and removed from the table as recommended by reviewer #2 (lines 251-253).

11. The proportion of reportable results for fresh plasma within the Burnett study is very low (77.1%). It would be useful to describe why this might have occurred. Further when indicating 'after re-testing', it would be useful to describe with which technologies (all or only plasma and DBS were re-tested).

R: We agree with your comment. We included more details about the retested specimens (lines 277-280). Furthermore, we included a description about the low proportion of plasma reportable results after first testing (lines 388-390).

12. The figures are currently presented as Figure 1, 3, and 2. This should be re-ordered in sequence of presentation in the results.

R: The figures were re-ordered in sequence of presentation in the results.

13. In the paragraph starting line 307, the proportion of reportable results started as 95.3% and

then decreased to 94.6%; however, this was presented as an increase. Please confirm the values.

R: This was corrected. There was a mistake on the calculation (line 333).

14. In Figure 3, it would be helpful to explain the straight diagonal line of results in each graph (some graphs have multiple). Further the y-axis could be improved (ie. what is 'DBS BPSD', I thought 'normal' DBS cards were used).

R: In Figure 3 (now Figure 2), the straight diagonal line corresponds to specimens with non-detectable VL results which were assigned fixed value of 1 to enable log10 transformation (lines 215-216). This was clarified in the figure legend. In fact, normal DBS cards were used for both Burnett and HemaSpot devices evaluations. The designation DBS BPSD and DBS HS was used to distinguish the Burnett (BPSD) and HemaSpot (HS) evaluations. This was clarified as well in the figure legend.

Minor comments

1. Line 19: 'provide' should be singular.

R: This was corrected (line 113)

2. Line 29: add 'the' before Burnett.

R: This was corrected (lines 3, 12,21)

3. Line 95 seems to have a footnote indicated, but no footnote.

R: This was corrected

4. Replace 'ART schemes' with 'ART regimens'.

R: This was corrected (line 88)

5. Line 106, replace 'measurement' with 'results'. Also here it would be useful to provide the high level results from those studies of the PSC.

R: This was corrected (line 99)

6. Line 111, remove 'device' after Burnet and make it plural on line 112. Similar on line 118, remove 'device' after Burnet.

R: This was corrected (lines 111,121)

7. Line 124, is the assay or lab ISO accredited?

R: We appreciate your comment. However, we would like to clarify that in our laboratory the ISO 15189 accreditation is for the VL assays and not for the laboratory (line 116-117). This laboratory performs many other assays (for example HIV sequencing) which are not accredited yet.

8. Line 135, 'the' Burnet device.

R: This was corrected (line 152)

9. Line 142, unclear what the 'SETM/HS' means - it doesn't seem to be included elsewhere.

R: This was corrected (line 101)

10. Line 280, I would suggest '498 patients had results for all four sample types: plasma...'

R: This was corrected (lines 298-299)

11. Similar to above, line 315 could say '550 patients had paired results for all specimens: plasma...'

R: This was corrected (lines 336-337)

12. Line 322, 'below' is spelled incorrectly.

R: This was corrected (line 343)

13. Line 322, instead of 'one' I would suggest writing out 'sensitivity' instead.

R: This was corrected (line 343,346)

14. In the figures, 'clot' is misspelled.

R: This was corrected

Reviewer's # 2 comments

General comment

Specify the HIV type of focus. HIV-1 or HIV-2

R: The entire manuscript was revised, and HIV was replaced by HIV-1.

Abstract

Background

General comment

Consider highlighting the disadvantages of using DBS in low VL

R: The comment was considered (lines 2-3 of the background)

Provide more context about the current state of ART and VL monitoring in Mozambique specifically, as this study is being conducted in that setting.

R: We agree with your comment. We incorporated data about ART and viral suppression in Mozambique (lines 69-72).

Line 95: Check the reference style and consider revising. One appears as a superscript.

R: This was corrected (line 81).

Study sites

Line 123-

124 indicates that the VL assays are ISO accredited instead of the reference lab. Kindly rephrase this statement as appropriate.

R: We appreciate your comment. However, we would like to clarify that the ISO 15189 accreditation in our laboratory is for the VL assays and not for the laboratory (line 116-117). This laboratory performs many other assays (for example HIV sequencing) which are not accredited yet.

HIV viral load testing

Line 211 mentions the cobas® Plasma Separation Card yet upto that point it does not appear in the methodology. Kindly clarify the relevance of its mention or inclusion.

R: The relevance of inclusion of cobas® Plasma Separation Card in this section was clarified (lines 208-212).

Data analysis

The section does not provide information on how the socio demographic and clinical information of the study population was analyzed. Kindly include.

R: A paragraph was incorporated to explain how the socio demographic and clinical information of the study was analyzed (lines 220-221).

Results

Line 237-

242 reports on analyzed socio demographic and clinical information yet the methodology section does not provide information on how these data were analyzed. Kindly revise.

R: A brief description of demographic and clinical data analysis was included (lines 220-221).

Table 1

The cumulative percentage for the Sex variable exceeds 100%.

R: This was corrected (table 1).

The Median (IQR) results when presented on the table are confusing, consider presenting them on a paragraph.

R: The Median (IQR) results were incorporated in two paragraphs (lines 251-253; 256-257).

Figure 1

The legend is unclear. For example “2 samples failed due to no sample”. Kindly rephrase.

R: This was clarified (lines 289-290).

Consider renumbering your figures on the text because figure 3 comes before figure 2.

R: This was corrected.

Line 286-290 represent significant findings. Kindly present the level of significance between the platforms using p values.

R: The Burnett and HemaSpot studies are independent evaluations which were conducted in different times, so we think that it is inappropriate to compare these devices. Our focus in this study was to compare each of the novel devices (Burnett and HemaSpot) separately against DBS, considering fresh plasma as reference. For this comparisons, sensitivity, specificity, misclassification rate and concordance correlation coefficient were calculated as described in the results section.

Table 2

The misclassification rate and false positive rate results of the HemaSpot are misrepresented. Kindly revise

R: This was corrected

Discussion

Line 364-365,367-368 indicate that a test platform was compared to a sample type. Kindly revise this section.

R: This was revised. We replaced the word “device” by “specimen type” (lines 396, 401)

Line 370-

372 mentions the advantages of the Burnett device while those of the HemaSpot device are not mentioned. Kindly revise.

R: This was revised (lines 400-407).

The limitations of the study are missing. Kindly revise.

R: This was revised. Limitations of the study were included (lines 427-429).

A lot of analytics have been done and presented. However, the meaning and interpretation of the finding has not been explained. Kindly consider revising

R: This was revised (lines 394-407).

Staff Comments:

Preparing Revision Guidelines

Please return the manuscript within 60 days; if you cannot complete the modification within this time period, please contact me. If you do not wish to modify the manuscript and prefer to submit it to another journal, please notify me of your decision immediately so that the manuscript may be formally withdrawn from consideration by Microbiology Spectrum.

August 25, 2023

Dr. Adolfo Vubil
Instituto Nacional de Saúde-Mozambique
Estrada Nacional número 1
Marracuene-Maputo
Mozambique

Re: Spectrum00546-23R1 (Performance of Two Plasma Separation Devices for HIV Viral Load Measurement in Primary Healthcare Settings)

Dear Dr. Adolfo Vubil:

Your manuscript has been accepted, and I am forwarding it to the ASM Journals Department for publication. You will be notified when your proofs are ready to be viewed.

Sincerely,

Yongjun Sui
Editor, Microbiology Spectrum
